# Immunomodulatory Effects of a High-CBD Cannabis Extract: A Comparative Analysis with Conventional Therapies for Oral Lichen Planus and Graft-Versus-Host Disease

**DOI:** 10.3390/ijms262110711

**Published:** 2025-11-03

**Authors:** Kifah Blal, Ronen Rosenblum, Hila Novak-Kotzer, Shiri Procaccia, Jawad Abu Tair, Nardy Casap, David Meiri, Ofra Benny

**Affiliations:** 1Department of Oral and Maxillofacial Surgery, Faculty of Dental Medicine, Hebrew University of Jerusalem, Jerusalem 9112102, Israel; dr.k.blal@mail.huji.ac.il (K.B.); jawadat@hadassah.org.il (J.A.T.); nardy.casap@mail.huji.ac.il (N.C.); 2School of Pharmacy, Faculty of Medicine, Hebrew University of Jerusalem, Jerusalem 9112001, Israel; 3Laboratory of Cancer Biology and Natural Drug Discovery, Faculty of Biology, Technion—Israel Institute of Technology, Haifa 3200003, Israel; rronen@campus.technion.ac.il (R.R.); nhila@technion.ac.il (H.N.-K.); shirip@technion.ac.il (S.P.)

**Keywords:** cannabis, cannabidiol, graft-versus-host disease, immunomodulation, oral lichen planus

## Abstract

This study investigates the immunomodulatory effects of a well-characterized cannabidiol (CBD)-rich cannabis extract, CAN296, on T lymphocytes (T cells), particularly Cluster of Differentiation 4 (CD4^+^) helper and Cluster of Differentiation 8 (CD8^+^) cytotoxic subsets, by examining T-cell activation, cytokine secretion, and cytotoxic molecule expression in comparison with the conventional treatments dexamethasone (DEX) and tacrolimus (TAC). It addresses key processes involved in the formation of premalignant immune-mediated lesions, such as those seen in oral lichen planus (OLP) and oral manifestations of graft-versus-host disease (oGVHD). CD4^+^ and CD8^+^ T cells were isolated from healthy donors and assessed in vitro for T cell activation via CD69 expression, secreted tumor necrosis factor alpha (TNF-α) and interferon gamma (IFN-γ) levels according to enzyme-linked immunosorbent assay (ELISA), and cytotoxic molecule expression Granzyme B, Perforin, Fas Ligand (Fas-L) quantified by flow cytometry. Cells were treated with different doses of CAN296 (2, 4, 8 µg/mL), DEX (0.4, 4, 40 µg/mL), or TAC (0.1, 1, 10 ng/mL), and all parameters were compared to untreated controls. CAN296 significantly inhibited T cell activation, reducing CD69 expression in CD4^+^ T cells to 2–11% and in CD8^+^ T cells to 5–17%. It also markedly suppressed TNF-α secretion in CD4^+^ T cells at all concentrations (*p* < 0.0001). In CD8^+^ T cells, CAN296 led to a near-complete reduction in TNF-α and IFN-γ, leaving both cytokines barely detectable at all tested doses (*p* < 0.0001). The effect of cell inhibition was significantly more pronounced than that observed with DEX or TAC, displaying dose-dependent reductions. TAC inconsistently lowered TNF-α while paradoxically increasing IFN-γ at lower concentrations. Additionally, CAN296 consistently suppressed cytotoxic molecule expression, reducing Granzyme B by 81–82%, Perforin by 40–53%, and Fas-L by 40–44%. DEX showed variable effects on cytotoxic molecule expression. At the same time, TAC demonstrated inconsistent modulation of Perforin and Granzyme B. Overall, CAN296 outperformed DEX and TAC, demonstrating more potent and consistent immunomodulatory effects. CBD-rich cannabis extract, CAN296, exhibits potent immunomodulatory properties by effectively inhibiting T cell activation, lowering pro-inflammatory cytokines, and suppressing cytotoxic molecule expression. Its efficacy surpasses conventional therapies like DEX and TAC, offering a promising novel treatment modality for T cell-mediated disorders, including OLP and oGVHD. These findings support further development of CAN296 formulations to optimize dosing and delivery, followed by clinical trials to validate its therapeutic potential.

## 1. Introduction

Oral lichen planus (OLP) is a chronic immune-mediated disorder affecting the oral mucosa, characterized by T cell-mediated destruction of basal keratinocytes [1]. While the etiology of OLP remains uncertain, an imbalance in cellular immunity is recognized as a key factor in disease onset and progression [2]. Clinically, OLP presents with reticular, atrophic, erosive, or ulcerative lesions that can cause significant morbidity [3]. While OLP is primarily an inflammatory condition, it is also classified as an Oral Potentially Malignant Disorder (OPMD) due to its association with an increased risk of malignant transformation ranging from 0.5% to 4% into oral squamous cell carcinoma (SCC) [4,5,6].

Similarly, graft-versus-host disease (GVHD) is a major complication of allogeneic Hematopoietic Stem Cell Transplantation (HSCT), where donor T cells mount an alloimmune response against host tissues. GVHD affects multiple organ systems and is associated with significant morbidity and mortality [7,8]. Oral manifestation of GVHD (oGVHD) is a considerable concern, presenting with lichenoid lesions that are clinically and histologically similar to OLP. Like OLP, oGVHD is associated with an increased risk of malignant transformation, further complicating disease management and prognosis [9,10].

In OLP and oGVHD, malignant transformation is driven by chronic inflammation, which promotes oxidative stress, DNA damage, and epithelial dysplasia [11]. Persistent immune activation increases the production of pro-inflammatory cytokines, including TNF-α and IFN-γ, leading to keratinocyte apoptosis and epithelial barrier disruption [12]. Additionally, the epithelium is directly attacked by Granzyme B and Perforin secreted by cytotoxic T lymphocytes [13], further exacerbating tissue destruction and contributing to disease progression.

The current standard of care for OLP and oGVHD consists of topical and systemic immunosuppressive agents, including corticosteroids such as dexamethasone (DEX) [14] and calcineurin inhibitors such as Tacrolimus (TAC) [15]. Corticosteroids exert their effects by binding to glucocorticoid receptors, suppressing pro-inflammatory cytokines such as TNF-α and IL-1β, inhibiting T cell proliferation, and inducing apoptosis in inflammatory cells [16,17]. Calcineurin inhibitors prevent T-cell activation by blocking the dephosphorylation of activated T-cells (NFAT) nuclear factor, thereby reducing IL-2 production and subsequent T-cell proliferation [18]. Given the chronic inflammatory nature and T-cell-driven pathology of OLP and oGVHD, the current treatments provide symptom relief rather than addressing the underlying mechanisms of immune dysregulation [19,20,21,22].

Alternative modalities such as topical or systemic retinoids, Photodynamic Therapy (PDT), and Low-Level Laser Therapy (LLLT) have been investigated to reduce corticosteroid dependence. Retinoids regulate epithelial differentiation but often cause mucosal irritation [23] and carry a well-established risk of teratogenicity [24]. PDT and LLLT can transiently reduce pain and erythema through reactive-oxygen–mediated immune modulation, yet outcomes are inconsistent and relapse rates are high [25,26,27]. Collectively, these modalities show variable efficacy and safety, underscoring the need for targeted, mechanism-based immunomodulators.

Although the immune-driven pathology of OLP and oGVHD is well-established, a major barrier to developing effective therapies is the lack of in vitro models that accurately replicate the immune-epithelial interface characteristic of these diseases. Existing systems, primarily based on oral keratinocyte monocultures, fail to capture the complex interactions between immune cells and epithelial tissue. As a result, many potential treatments cannot be adequately evaluated for disease-specific mechanisms. To address this gap, our study focuses directly on key lymphocyte-mediated processes, including CD4^+^ and CD8^+^ T cell activation, TNF-α and IFN-γ secretion, and the expression of cytotoxic mediators such as Perforin and Granzyme B. These core immune markers are relevant to both OLP and oGVHD. This approach provides a targeted and mechanistically relevant assessment of immunomodulation in the absence of reliable disease models.

Cannabis-derived extracts, particularly CBD, the main non-psychoactive constituent, exhibit potent immunomodulatory [28,29,30] and anticancer properties [31,32]. They exert their effects through interactions within the Endocannabinoid System (ECS) [33] via modulation of Cannabinoid Receptor Type 2 (CB2), Transient Receptor Potential (TRP), and Peroxisome Proliferator-Activated Receptor (PPAR) pathways [34,35,36], as well as through inhibition of the Nuclear Factor κB (NF-κB) signaling cascade, leading to suppression of T-cell activation, reduced secretion of pro-inflammatory cytokines such as TNF-α and IFN-γ [37,38,39], and decreased expression of cytotoxic molecules [28,40]. These mechanisms make them promising therapeutic candidates for immune-mediated oral diseases [41,42]. Despite these promising findings, well-controlled clinical studies assessing the efficacy and long-term safety of cannabinoid-based therapies in OLP and oGVHD remain limited, underscoring the need for further translational and clinical research.

Given the shared immune dysregulation, chronic inflammation, and oncogenic risk in OLP and oGVHD, novel therapeutic strategies are required to address both disease progression and malignant transformation risk. CBD’s immunomodulatory effects may mitigate T cell-mediated tissue damage by reducing cytokine secretion and suppressing cytotoxic molecule expression. Its interaction with the ECS and other immune-regulatory pathways [33,43] could help control inflammation and prevent malignant transformation. Additionally, cannabinoids’ ability to induce apoptosis in dysregulated immune cells [39,44] suggests the potential to restore immune homeostasis without broadly suppressing immune function.

This study explored the immunomodulatory effects of CBD-rich cannabis extract compared to conventional therapies, focusing on key immune mechanisms such as T-cell activation, cytokine secretion, and cytotoxic molecule expression. By investigating CBD’s ability to regulate immune responses without broadly suppressing immunity, this research aims to provide critical insights into its therapeutic potential in OLP and oGVHD and explore whether CBD-based therapies could represent a safer, more targeted alternative to current immunosuppressive treatments.

### Study Hypothesis

The null hypothesis of this study states that the CBD-rich cannabis extract CAN296 does not exert significant immunomodulatory effects on T-cell activation, cytokine secretion, or cytotoxic molecule expression compared with the conventional immunosuppressants dexamethasone and tacrolimus. The alternative hypothesis proposes that CAN296 modulates these immune parameters in a dose-dependent manner, providing a more targeted immunomodulatory profile relevant to OLP and oGVHD.

## 2. Results

### 2.1. CAN296 and CBD:CBC (2:1) Combination Significantly Inhibited TNF-α and IFN-γ Secretion by CD8^+^ T Cells

Our previous study demonstrated the anti-proliferative and pro-apoptotic effects of the CAN296 extract and the CBD:CBC combination on head and neck squamous cell carcinoma cells [31]. In the current work, we aim to explore the immunomodulatory potential of CAN296 extract and the CBD:CBC combination on these cancerous cells. We first evaluated their ability to modulate the secretion of the pro-inflammatory cytokines TNF-α and IFN-γ by CD8^+^ T cells, as these play key roles in T cell-mediated immune responses. CD8^+^ T cells were treated with CAN296 or CBD:CBC at 4 µg/mL and 8 µg/mL, and cytokine secretion was quantified.

Baseline secretion levels in the control group averaged 1434.5 pg/mL for TNF-α and 897.2 pg/mL for IFN-γ. For TNF-α secretion (Figure 1A), treatment with CAN296 led to a dose-dependent reduction, with a 51% and 80% decrease at 4 µg/mL and 8 µg/mL, respectively, relative to the control. Similarly, the CBD:CBC combination reduced TNF-α secretion by 27% at 4 µg/mL and 76% at 8 µg/mL. For IFN-γ secretion (Figure 1B), CAN296 reduced secretion by 58% at 4 µg/mL and 97% at 8 µg/mL, while the CBD:CBC combination led to reductions of 21% at 4 µg/mL and 91% at 8 µg/mL.

Given that both CAN296 and the CBD:CBC combination effectively suppressed cytokine secretion, we chose to focus on CAN296 for the remainder of this study due to its consistent effects and its status as a whole extract already approved for medical use, whereas the CBD:CBC formulation would require additional testing and regulatory approval before clinical application. This decision aligned with our prior work, demonstrating CAN296’s efficacy in head and neck squamous cell carcinoma models. Thus, subsequent experiments compared CAN296 to conventional immunosuppressants (DEX and TAC).

### 2.2. CAN296 Suppresses CD69 Expression on CD4^+^ and CD8^+^ T Cells, Outperforms Dexamethasone and Tacrolimus

To evaluate the effects of CAN296 on T cell activation in both CD4^+^ helper and CD8^+^ cytotoxic subsets, we compared its efficacy to DEX and TAC, using concentrations reported in previous studies [18,45]. The following concentrations were used: CAN296 at 2, 4, and 8 µg/mL, DEX at 0.4, 4, and 40 µg/mL, and TAC at 0.1, 1, and 10 ng/mL. CD4^+^ and CD8^+^ T cell activation was assessed by measuring CD69 expression 24 h post-treatment.

Flow cytometry analysis (Figure 2A–C), normalized to 100% in untreated controls, revealed that CAN296 significantly reduced CD69 expression in CD8^+^ T cells in a dose-dependent manner (*p* < 0.01). At low, medium, and high doses, CAN296-treated cells showed 17%, 5%, and 5% CD69+ cells, respectively, whereas TAC yielded 124%, 84%, and 52%, and DEX resulted in 71%, 74%, and 55%. A similar pattern was observed in CD4^+^ T cells (Figure 2D–F). With untreated cells set to 100%, CAN296 lowered CD69 expression to 11%, 2%, and 2% at its low, medium, and high doses, while TAC led to 99%, 49%, and 42%, and DEX produced 140%, 139%, and 50%. Two-way ANOVA (Figure 2G) confirmed a significant effect of treatment type (*p* < 0.0001), with no notable difference between CD4^+^ and CD8^+^ T cells (*p* = 0.5205) and no interaction effects (*p* = 0.0937). These findings demonstrate that CAN296 effectively reduces T cell activation in both subsets, whereas DEX and TAC exhibit more variable outcomes.

### 2.3. CAN296 Inhibits TNF-α and IFN-γ Secretion by CD4^+^ and CD8^+^ T Cells, Outperforming Dexamethasone and Tacrolimus

To assess the effects of CAN296, TAC, and DEX on pro-inflammatory cytokine secretion, TNF-α and IFN-γ levels in the supernatants of treated CD4^+^ and CD8^+^ T cells were measured by ELISA after 24 h of treatment.

In CD4^+^ T cells, CAN296 significantly reduced TNF-α secretion at all concentrations (*p* < 0.0001). TAC and DEX significantly reduced TNF-α secretion at medium and high concentrations (*p* < 0.0001). In contrast, at the low concentration, both showed a paradoxical slight increase in cytokine secretion compared to the control (Figure 3A). Similar trends were observed for IFN-γ in CD4^+^ T cells across all treatments and doses; however, these changes did not reach statistical significance (Figure 3B). Two-way ANOVA confirmed that treatment type and dose significantly influenced TNF-α secretion (*p* < 0.0001), whereas IFN-γ levels remained unaffected.

In CD8^+^ T cells, CAN296 was highly effective, resulting in significant reductions in TNF-α and IFN-γ secretion at all concentrations (*p* < 0.0001) (Figure 3C,D). DEX also significantly reduced TNF-α at low and medium-high concentrations (*p* = 0.001, *p* < 0.0001), while IFN-γ was decreased only at the high concentration (*p* < 0.0001). TAC significantly reduced TNF-α at medium and high concentrations (*p* < 0.0001) and IFN-γ at medium and high concentrations (*p* = 0.01, *p* < 0.001) (Figure 3C,D). When comparing the percentage decrease across different concentrations within each treatment group, CAN296’s efficacy remained consistently high, whereas DEX and TAC showed more pronounced concentration-dependent effects. For TNF-α, TAC exhibited significant differences between low vs. medium/high (*p* < 0.0001) and between medium vs. high (*p* = 0.0007). At the same time, DEX showed substantial differences between low vs. medium (*p* = 0.0130) and between low/medium vs. high (*p* < 0.0001) (Figure 3E). For IFN-γ, TAC displayed significant differences between low vs. medium/high (*p* < 0.0001), and DEX showed significant changes between low vs. medium (*p* = 0.0002) and low/medium vs. high (*p* < 0.0001) (Figure 3F). Two-way ANOVA revealed significant effects of interaction (*p* < 0.0001), row factor (*p* < 0.0001), and column factor (*p* < 0.0001) for both cytokines in CD8^+^ T cells, indicating that treatment type and concentration collectively influenced TNF-α and IFN-γ secretion.

### 2.4. CAN296 Reduces Fas-L Expression in CD8^+^ T Cells, Outperforming Dexamethasone and Tacrolimus

To evaluate the effects of CAN296 on Fas-L-mediated T cell cytotoxicity and compare it to DEX and TAC, Fas-L expression in activated CD8^+^ T cells was examined after 24 h of treatment.

Flow cytometry analysis (Figure 4A–C) showed that Fas-L expression levels decreased in response to CAN296 and DEX across all concentrations, whereas TAC treatment elevated Fas-L levels at higher doses. Two-way ANOVA confirmed that treatment type and concentration significantly influenced Fas-L expression (*** *p* < 0.0001).

### 2.5. CAN296 Inhibits Granzyme B and Perforin Expression in CD8^+^ T Cells, Comparable to Dexamethasone and Tacrolimus

To evaluate the effects of CAN296 on CD8^+^ T cell cytotoxicity, we measured the expression of Granzyme B and Perforin, two key mediators of T cell-induced apoptosis. We compared the results to those of DEX and TAC.

CD8^+^ T cells were treated for 48 h with each agent, and protein expression was analyzed by flow cytometry (Figure 5A–F). CAN296 significantly reduced Perforin expression, lowering levels to 47%, 60%, and 51% at low, medium, and high doses, respectively (equating to 40–53% reductions vs. control) (Figure 5G). Granzyme B was markedly reduced to 19%, 22%, and 19% at the same doses, representing 78–81% decreases. TAC showed similar suppression of Perforin, lowering it to 41% at low dose and 45% at medium/high, while Granzyme B was consistently reduced to 9% across all TAC doses (~91% decrease). DEX displayed a more variable response: Perforin dropped to 63%, 35%, and 63% at low, medium, and high doses, respectively, whereas Granzyme B levels fell to 19%, 7%, and 13%, representing 81–93% reductions. Two-way ANOVA revealed significant effects of treatment type and concentration (**** *p* < 0.0001), confirming that all three agents modulated Perforin and Granzyme B expression. Overall, these findings indicate that CAN296 effectively suppresses key cytotoxic markers in CD8^+^ T cells—on par with TAC and DEX—and remains consistently potent even at lower doses, supporting its potential therapeutic application in immune-mediated diseases such as OLP and oGVHD.

## 3. Discussion

This study is our team’s second publication on the high-CBD strain CAN296. While our previous research demonstrated its cytotoxic effects against head and neck squamous cell carcinoma (HNSCC) cells, the present study explores the immunomodulatory potential of CAN296 as a potential candidate for treating immune-mediated oral disorders, particularly oral lichen planus (OLP) and oral manifestations of graft-versus-host disease (oGVHD).

This study aims to evaluate the immunomodulatory effects of CAN296 and a CBD:CBC combination (2:1), as used in our earlier study [31]. The goal was to assess their ability to modulate the secretion of pro-inflammatory cytokines (TNF-α and IFN-γ) by CD8^+^ T cells, which play key roles in cell-mediated immune responses in premalignant lesions such as OLP and oGVHD. After confirming that both CAN296 and CBD:CBC treatments significantly reduced cytokine secretion, we further examined CAN296 because of its consistent in vitro effects and its availability as a whole cannabis extract already approved for medical use. Since CAN296 previously showed cytotoxic activity in HNSCC cell models, we aimed to evaluate whether it also comprises effects on the immune system compared with conventional immunosuppressants, DEX, and TAC.

Although the immune-driven pathology of OLP and oGVHD is well-characterized, a significant barrier to developing effective treatments lies in the absence of reliable in vitro models that accurately replicate the immune-epithelial dynamics of these diseases. Most current systems rely on oral keratinocyte monocultures and fail to capture the complex bidirectional signaling between immune cells and epithelial tissue. This limitation hampers the ability to evaluate immune-modulating therapies in a disease-relevant context. To overcome this, our study focused directly on the immune component by assessing CD4^+^ and CD8^+^ T cell activation, TNF-α and IFN-γ secretion, and expression of cytotoxic molecules such as Perforin and Granzyme B. These immune markers are central to the pathogenesis of both OLP and oGVHD and provide a targeted framework for assessing the immunomodulatory potential of candidate therapies like CAN296.

Cannabis extracts have been reported to possess immunosuppressive properties [38,39,46]. However, few studies have compared their activity with that of commonly used immunosuppressants. The present work contributes to this knowledge by examining the effects of a cannabis extract on key immune mechanisms in premalignant immune-mediated lesions, including T-cell activation, cytokine secretion, and cytotoxic molecule expression. These processes are central to immune-mediated lesions such as OLP and oGVHD, which are associated with an increased risk of malignant transformation to HNSCC.

CAN296 demonstrated substantial suppression of T cell activation, as evidenced by reduced CD69 expression in both arms of the T lymphocyte response, immune modulation (CD4^+^), and cytotoxicity (CD8^+^). CAN296 treatment reduced CD69 expression in CD4^+^ T cells consistently at all doses. In comparison, DEX decreased CD69 expression only at high concentrations, whereas TAC showed a dose-dependent but inconsistent reduction. This reduction is consistent with previous reports showing that CBD, the main constituent of CAN296, can modulate T-cell activity [33,40].

While CAN296 significantly suppressed TNF-α secretion in CD4^+^ T cells, its effects on IFN-γ were insignificant, which can be attributed to donor-to-donor variability and the limited sample size in this study. Larger-scale studies will be necessary to determine whether CAN296 consistently modulates IFN-γ secretion in CD4^+^ T cells under more controlled and reproducible conditions.

Beyond T cell activation, CAN296 significantly reduced pro-inflammatory cytokine secretion, with TNF-α and IFN-γ levels decreasing by up to 96% in CD8^+^ T cells. In CD4^+^ T cells, CAN296 reduced TNF-α secretion across all concentrations but did not significantly alter IFN-γ levels. Previous research indicates that CBD inhibits NF-κB signaling and suppresses pro-inflammatory cytokines [42,47], and our findings here reinforce its potential role as an immunomodulator in immune-mediated diseases. In contrast, DEX and TAC only exhibited significant cytokine secretion suppression at higher concentrations, with TAC paradoxically increasing TNF-α at lower concentrations and DEX showing minimal reduction at lower doses. Such paradoxical effects at low doses have been reported under specific conditions [48], but this phenomenon is not the focus of our research and has not been further pursued. This variability suggests that CAN296 provides a stable and effective approach to modulating cytokine responses in CD4^+^ and CD8^+^ T cells. These cytokines are central to the pathogenesis and progression of immune-mediated lesions, driving immune cell recruitment, activation, cytotoxicity, barrier disruption, inflammation, tissue damage, and malignant transformation in conditions such as OLP and oGVHD lesion development [1,49].

In addition to its effects on cytokine secretion, CAN296 demonstrated robust suppression of key cytotoxic mediators associated with T cell-induced epithelial damage. Specifically, it effectively reduced the expression of Perforin and Granzyme B in CD8^+^ T cells—proteins directly involved in apoptotic signaling and tissue injury in immune-mediated lesions. Notably, CAN296 lowered Fas-L expression, a critical component of the extrinsic apoptotic pathway implicated in keratinocyte apoptosis in OLP and oGVHD. While conventional immunosuppressants like DEX and TAC showed similar reductions in Granzyme B and Perforin, they were less consistent in modulating Fas-L expression, with TAC paradoxically increasing it at higher doses—a phenomenon previously observed in immune cells treated with calcineurin inhibitors [49]. The ability of CAN296 to suppress both intracellular cytotoxic molecules and Fas-L suggests a potential to reduce T-cell–mediated epithelial injury in immune-driven oral diseases.

These cytotoxic pathways play a central role in T cell-mediated epithelial damage in immune-mediated lesions like OLP and oGVHD. CD8^+^ T cells release Perforin to create membrane pores in target keratinocytes, allowing Granzyme B to enter and activate caspases, leading to apoptosis. Fas-L binding to Fas (CD95) on keratinocytes also triggers the extrinsic apoptotic pathway, further driving cell death. These mechanisms contribute to keratinocyte apoptosis, epithelial barrier disruption, and lesion formation. The ability of CAN296 to consistently suppress these cytotoxic mediators, in contrast to the variable effects observed with DEX and TAC, highlights its potential to mitigate T cell-mediated tissue damage in immune-mediated diseases like OLP and oGVHD [19,20]. This suppression is attributed to CAN296’s interaction with multiple immune-modulating pathways [34,35], with CBD, the main active component in CAN296, shown to induce apoptosis in activated immune cells while sparing healthy tissues, which could explain its efficacy in chronic inflammatory conditions [39,44].

Study limitations acknowledgment. The experiments were conducted using CD4^+^ and CD8^+^ T cells isolated from nine healthy donors to establish a controlled immunological baseline for pharmacodynamic comparisons with DEX and TAC. While this approach ensured reproducibility, it may not fully capture the immune dysregulation characteristic of OLP and oGVHD. Each assay was performed using samples from 2–4 of these donors, which may have contributed to inter-donor variability and limited statistical power.

Although direct viability assays were not repeated in this study, all concentrations were within validated non-toxic ranges established in our previous work [31] and supported by inclusion of a vehicle-only control (0.1% DMSO), confirming that solvent exposure had no detectable immunological effects Furthermore, the specific molecular pathways through which CBD-type cannabis extracts exert their immunomodulatory effects were beyond the scope of the present work. Future investigations should therefore include a larger donor cohort, patient-derived T cells, and advanced co-culture or organotypic oral-mucosa models to explore further and address these limitations.

While these findings are based on in vitro experiments with a limited number of donors, the well-documented therapeutic applications of cannabis extracts in conditions such as chronic pain, epilepsy, and multiple sclerosis provide a strong foundation for further exploring the potential of CAN296 in treating immune-mediated diseases such as OLP and oGVHD. As CAN296 is an already approved cannabis extract for medical use, further research should focus on optimizing its application for these conditions. Future studies should develop standardized formulations of CAN296 and determine optimal dosing strategies. Once these parameters are established, human in vivo trials should be conducted to evaluate efficacy, safety, and tolerability, particularly for chronic and relapsing conditions such as OLP and oGVHD. The immunomodulatory findings obtained with CAN296, together with the growing body of evidence on cannabinoid-based research, support its consideration for further preclinical and translational development.

## 4. Materials and Methods

### 4.1. Immunosuppressive Agents’ Preparation

TAC and DEX were obtained from MedChemExpress (Monmouth Junction, NJ, USA), prepared as stock solutions in dimethyl sulfoxide (DMSO; Sigma-Aldrich, St. Louis, MO, USA), and then stored at −20 °C. For working concentrations, the agents were diluted in Peripheral Blood Mononuclear Cell (PBMC) Medium. TAC was diluted to final concentrations of 0.1, 1, and 10 ng/mL. These concentrations were chosen to represent a wide therapeutic window, encompassing sub-therapeutic, therapeutic, and supra-therapeutic levels, as the IC_50_ for NFAT dephosphorylation and T cell inhibition is approximately 1–3 ng/mL [18]. DEX was diluted to final concentrations of 0.4, 4, and 40 µg/mL. These doses correspond to the range used in previous studies for evaluating dexamethasone’s impact on T cell activation and cytokine suppression, spanning concentrations below, near, and above the reported IC_50_ of approximately 4 µg/mL for inhibiting T cell proliferation and cytokine secretion [45]. To ensure the biological relevance of the selected concentrations, all DEX and TAC doses were chosen based on literature-reported IC_50_ values for T-cell inhibition and cytokine suppression [18,45] and were below known cytotoxic thresholds for human lymphocytes. Each treatment included a corresponding vehicle control containing 0.1% DMSO, which did not affect cytokine secretion or activation marker expression relative to untreated controls.

### 4.2. Phytocannabinoid Extraction and Sample Preparation

Air-dried Type III high-CBD cannabis inflorescences were obtained from an Israeli medical cannabis distributor. The inflorescences were extracted with ethanol and heat-decarboxylated at 130 °C (CAN296) [50]. CBD and Cannabichromene (CBC) were kindly provided by Prof. Rafael Mechoulam (Institute for Drug Research, Medical Faculty, Hebrew University, Jerusalem, Israel) and mixed in a 2:1 weight ratio. The compounds were dissolved in Dimethyl Sulfoxide (DMSO) (Sigma-Aldrich, St. Louis, MO, USA) and stored at −20 °C. For phytocannabinoid profiling, extracts were analyzed using a Thermo Scientific ultra-high-performance liquid chromatography (UHPLC) system coupled with a Q Exactive™ Focus Hybrid Quadrupole-Orbitrap mass spectrometer (Thermo Scientific, Bremen, Germany), as previously described [31]. Phytocannabinoid identification and absolute quantification were performed using analytical phytocannabinoid standards (Thermo Scientific, Bremen, Germany) and external calibrations, as previously described [51]. For working concentrations, extracts were diluted in PBMC Medium to achieve final concentrations of 2, 4, and 8 µg/mL. The final DMSO concentration in cell culture experiments did not exceed 0.1% to avoid cytotoxicity. To exclude potential vehicle-related effects, control wells containing 0.1% DMSO without active compounds were included in all experiments. No measurable differences were detected between vehicle-treated and untreated controls in cytokine secretion or flow cytometry parameters. The concentrations of CAN296 (2–8 µg/mL) were selected within biologically relevant ranges based on our previous study [31] and related reports evaluating cannabinoid immunomodulatory activity [44,52,53]. These doses were verified in earlier experiments to be non-cytotoxic and to maintain T-cell viability above 95%.

### 4.3. PBMC Isolation and CD4^+^/CD8^+^ Enrichment

Peripheral blood mononuclear cells (PBMCs) were isolated from venous blood collected from 9 healthy donors at Sheba Tel Hashomer University Hospital using 6 mL EDTA anticoagulant tubes (BD Vacutainer^®^, BD Biosciences, Franklin Lakes, NJ, USA). Blood samples were diluted 1:1 with sterile Phosphate-Buffered Saline (PBS; Gibco, Thermo Fisher Scientific, Waltham, MA, USA) and separated at room temperature by density gradient centrifugation using Ficoll-Paque PLUS (Cytiva, Marlborough, MA, USA; formerly GE Healthcare Life Sciences). The buffy coat layer containing PBMCs was carefully collected and washed twice with PBS.

For lymphocyte enrichment, PBMCs were incubated with either RosetteSep™ Human CD4^+^ or CD8^+^ T Cell Enrichment Cocktail (StemCell Technologies, Vancouver, BC, Canada) following the manufacturer’s instructions. Enriched CD4^+^ and CD8^+^ T cells were resuspended in PBMC Medium at 1 × 10^6^ cells/mL.

PBMCs were maintained in RPMI-1640 medium (Gibco, Thermo Fisher Scientific, Waltham, MA, USA) supplemented with 10% Fetal Bovine Serum (FBS; heat-inactivated, Gibco, Thermo Fisher Scientific, Waltham, MA, USA), 1% penicillin-streptomycin (100 U/mL penicillin, 100 µg/mL streptomycin; Gibco, Thermo Fisher Scientific, Waltham, MA, USA), and 2 mM L-glutamine (Gibco, Thermo Fisher Scientific, Waltham, MA, USA).

### 4.4. T Cell Activation

CD4^+^ and CD8^+^ T cells were activated using anti-CD3 and anti-CD28 antibodies (BioLegend, San Diego, CA, USA), coated onto flat-bottom 96-well plates at final concentrations of 5 µg/mL and 2 µg/mL, respectively. Plates were incubated at 4 °C for 1 h to allow for antibody adhesion, and then washed before adding isolated T cells resuspended in PBMC Medium at 1 × 10^5^ cells per well. Cells were incubated at 37 °C with 5% CO_2_ for 24–48 h in a humidified incubator. Following incubation, supernatants were collected for cytokine analysis by Enzyme-linked immunosorbent assay (ELISA), and cells were harvested for flow cytometry analysis of activation markers.

### 4.5. CD69, Granzyme B, Perforin, and Fas-L Staining

The CD69 activation marker was assessed on CD4^+^ and CD8^+^ T cells following 24 h activation, while Granzyme B, Perforin, and Fas-L were evaluated in CD8^+^ T cells after 48 h. For surface staining, cells were incubated with APC-conjugated anti-human CD69 or APC-conjugated anti-human Fas-L antibodies (BioLegend, San Diego, CA, USA) or respective isotype controls for 30 min at 4 °C in the dark. For intracellular staining, cells were fixed with 4% paraformaldehyde (PFA) at room temperature for 15 min, washed with Phosphate Buffered Saline (PBS), and permeabilized using BD Perm/Wash™ Buffer (BD Biosciences, Franklin Lakes, NJ, USA) following the manufacturer’s instructions. Permeabilized cells were incubated with FITC-conjugated anti-human/mouse Granzyme B and Brilliant Violet 421™-conjugated anti-human Perforin antibodies (BioLegend, San Diego, CA, USA) for 30 min at 4 °C in the dark. FITC- and BV421-conjugated isotype controls were included. Stained cells were washed with PBS containing 2% FBS, resuspended in PBS, and analyzed by flow cytometry.

### 4.6. Flow Cytometry Analysis

Stained cells were analyzed using a BD LSRFortessa™ flow cytometer (BD Biosciences, Franklin Lakes, NJ, USA). Data acquisition included 10,000 events per sample, with compensation performed using single-stained controls for each fluorochrome. Fluorescence-minus-one (FMO) and isotype controls were included to establish gating thresholds and assess non-specific binding. Data processing was performed using FlowJo software (version 10.8.1; Tree Star, Inc., Ashland, OR, USA). Lymphocytes were gated based on the properties of forward scatter (FSC) and side scatter (SSC). Geometric mean fluorescence intensity (MFI), median fluorescence intensity, and the percentage of positive cells were calculated for each marker. The results were expressed as the percentage of positive cells within the CD8^+^ T cell population and MFI values where applicable.

### 4.7. TNF-α and IFN-γ Secretion Analysis by ELISA

TNF-α and IFN-γ levels were quantified using BioSource ELISA Kits (Thermo Fisher Scientific, Waltham, MA, USA). Supernatants were collected from CD4^+^ or CD8^+^ T cells after 24 h treatment. Standards were serially diluted to generate calibration curves, and blanks containing assay diluent were included in the 96-well plate. Plates were incubated with 50 µL of biotin-conjugated detection antibody at room temperature (2 h for TNF-α; 1.5 h for IFN-γ), washed, and then incubated with 100 µL of streptavidin-HRP conjugate for 30–45 min in the dark. The substrate reaction was initiated by adding 100 µL of TMB for 30 min, followed by 100 µL of stop solution. Absorbance was measured at 450 nm using a microplate reader (BioTek, Agilent Technologies, Santa Clara, CA, USA). Cytokine concentrations were determined by interpolating absorbance values against the standard curve, which was confirmed to be linear. All samples were run in duplicate to ensure reproducibility.

### 4.8. Statistical Analysis

Statistical analyses were performed using GraphPad Prism (version 9.3.1; GraphPad Software, LLC, Boston, MA, USA). The results were expressed as the mean ± SEM from at least three independent experiments. In some analyses, data were normalized and presented as a percentage of the control. Comparisons between multiple groups were conducted using one-way or two-way ANOVA, followed by Bonferroni’s, Šidák’s, or Dunnett’s multiple comparisons tests, as indicated. Statistical significance was set at *p* ≤ 0.05 for all analyses.

### 4.9. Study Approval

The research protocols, including donor recruitment, sample collection, and data handling, were reviewed and approved by the Institutional Ethics Committee of the Technion–Israel Institute of Technology (Haifa, Israel). Magen David Adom (MDA; Israeli National Blood Bank, Tel Aviv, Israel) randomly collected and provided human peripheral blood samples with written informed consent obtained from all donors. Donor identities remained anonymous throughout this study.

## 5. Conclusions

This study demonstrates that the CBD-rich cannabis extract CAN296 effectively suppresses CD4^+^ helper and CD8^+^ cytotoxic T cells by reducing activation, pro-inflammatory cytokine secretion, and cytotoxic molecule expression. By modulating both arms of the immune response, key drivers of immune-mediated lesions such as OLP and oGVHD, CAN296 outperformed conventional therapies like DEX and TAC, even at low concentrations. These findings position CAN296 as a promising targeted immunomodulatory agent with the potential to reduce tissue damage and prevent lesion formation and premalignant progression. Future research should focus on developing optimized oral formulations to enhance bioavailability, stability, and localized delivery in the oral cavity.

## Figures and Tables

**Figure 1 ijms-26-10711-f001:**
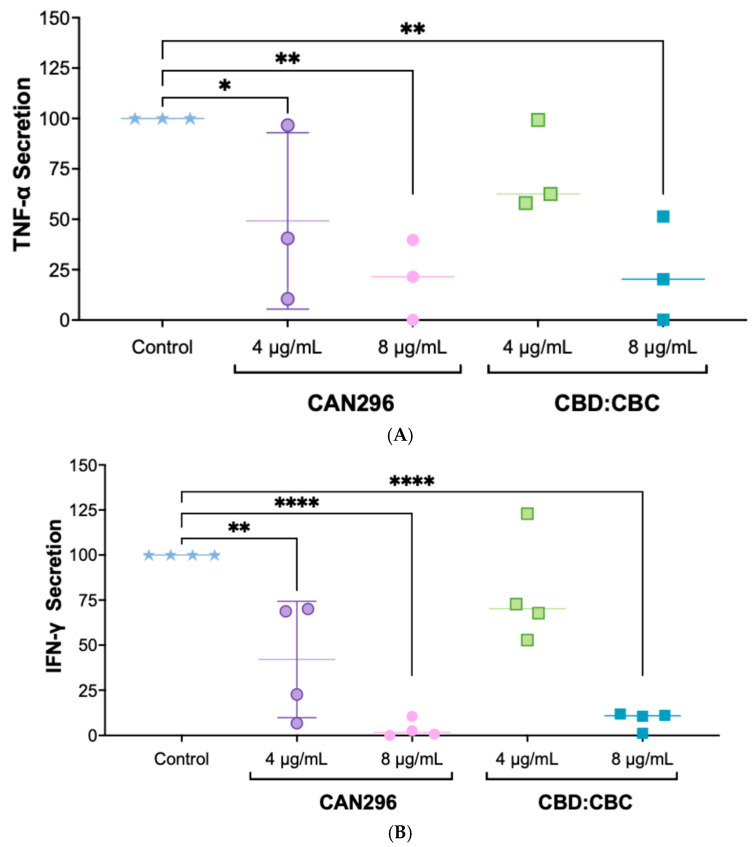
Effects of CAN296 and CBD:CBC (2:1) on TNF-α and IFN-γ secretion by CD8^+^ T cells. CD8^+^ T cells from (*n* = 3 donors) (TNF-α) and (*n* = 4 donors) (IFN-γ) were treated with CAN296 or CBD:CBC (4 and 8 µg/mL; 2:1 ratio) for 24 h. Untreated cells served as controls. (**A**) TNF-α and (**B**) IFN-γ secretion are shown as percentage changes relative to controls. Data represent mean ± SE (standard error). Statistical significance was determined using two-way ANOVA with Dunnett’s post hoc test (*p* = 0.0174, * *p* = 0.0012 and 0.0016 for TNF-α; ** *p* = 0.0020, **** *p* < 0.0001 for IFN-γ). Abbreviations: TNF-α, tumor necrosis factor alpha; IFN-γ, interferon gamma; CBD, cannabidiol; CD8^+^, Cluster of Differentiation 8 cytotoxic T cells; CBC, cannabichromene; SE, standard error; ANOVA, analysis of variance.

**Figure 2 ijms-26-10711-f002:**
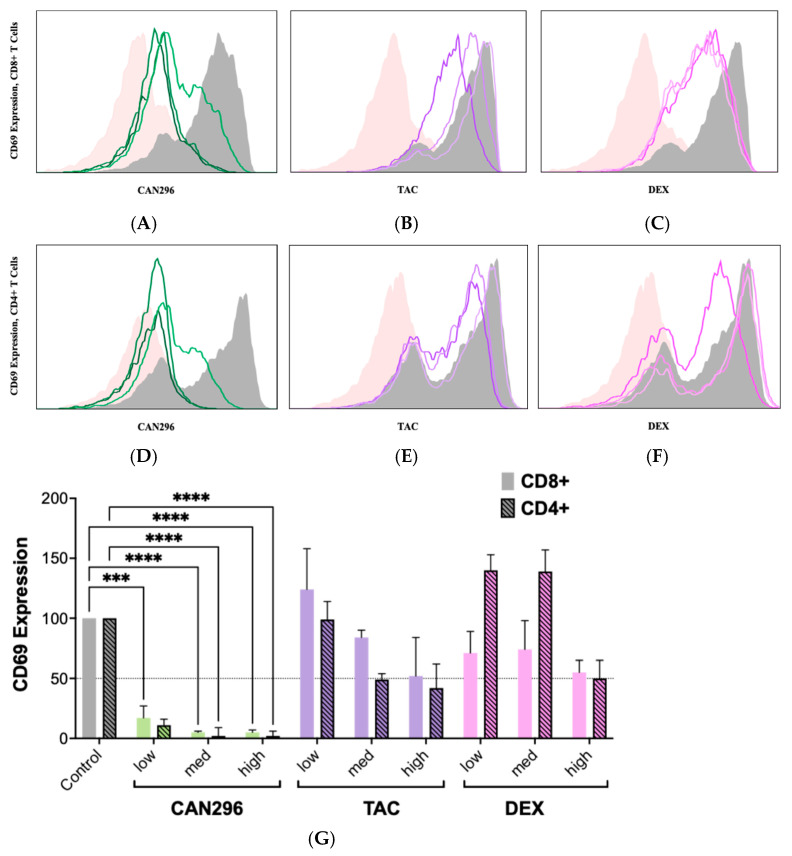
Effects of CAN296, TAC, and DEX on CD69 expression in CD4^+^ and CD8^+^ T cells. Representative histograms show CD69 expression in CD8^+^ (**A**–**C**) and CD4^+^ (**D**–**F**) T cells (*n* = 2 donors) after 24 h treatment with CAN296 (2–8 µg/mL), TAC (0.1–10 ng/mL), or DEX (0.4–40 µg/mL). Panels (**A**,**D**) (green) show CAN296, (**B**,**E**) (purple) TAC, and (**C**,**F**) (red) DEX; darker shades indicate higher concentrations. Gray lines denote positive controls (stimulated, untreated), and pink lines represent negative controls (unstimulated). (**G**) CD69 expression levels (MFI, normalized to control). The dashed horizontal line marks 50% of the normalized control value, serving as a visual reference for expression reduction. Data are presented as medians with 95% confidence intervals. Statistical significance was determined using two-way ANOVA with Šidák’s multiple comparisons test (*** *p* = 0.0003, **** *p* < 0.0001). Abbreviations: CD4^+^, Cluster of Differentiation 4 helper T cells; CD8^+^, Cluster of Differentiation 8 cytotoxic T cells; TAC, tacrolimus; DEX, dexamethasone; MFI, median fluorescence intensity; ANOVA, analysis of variance.

**Figure 3 ijms-26-10711-f003:**
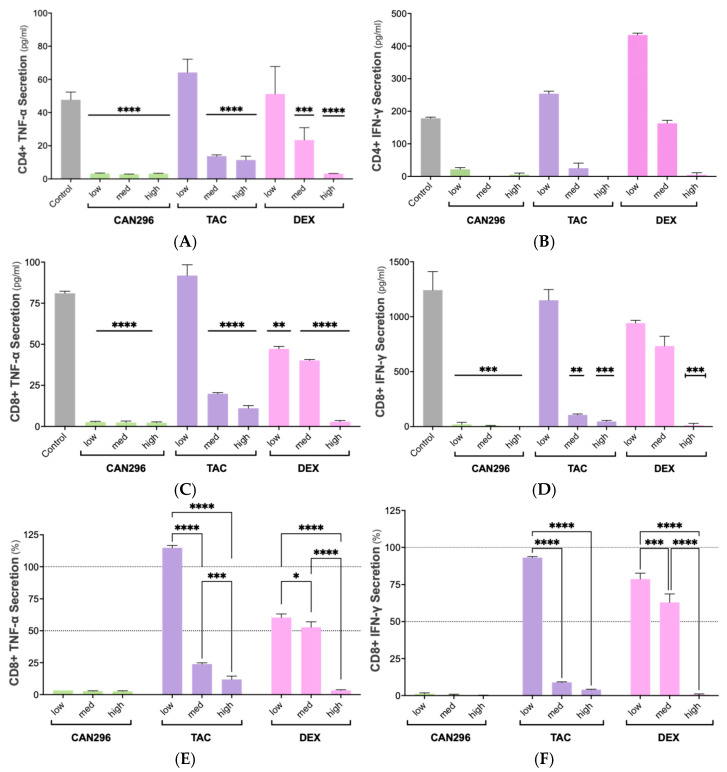
Effects of CAN296, DEX, and TAC on TNF-α and IFN-γ secretion by CD4^+^ and CD8^+^ T cells. Cells from (*n* = 2 donors) were treated with CAN296 (2–8 µg/mL), DEX (0.4–40 µg/mL), or TAC (0.1–10 ng/mL) for 24 h. (**A**,**B**) TNF-α and IFN-γ levels (pg/mL) in CD4^+^ T cells; (**C**,**D**) TNF-α and IFN-γ levels in CD8^+^ T cells; (**E**,**F**) percentage decrease relative to untreated controls. The dashed horizontal lines mark 50% and 100% of the normalized control values, serving as a visual reference thresholds for cytokine secretion. Data are mean ± SD. Statistical significance was determined using Dunnett’s multiple comparisons test (* *p* < 0.05, ** *p* < 0.01, *** *p* < 0.001, **** *p* < 0.0001). Abbreviations: CD4^+^, Cluster of Differentiation 4 helper T cells; CD8^+^, Cluster of Differentiation 8 cytotoxic T cells; DEX, dexamethasone; TAC, tacrolimus; TNF-α, tumor necrosis factor alpha; IFN-γ, interferon gamma; SD, standard deviation.

**Figure 4 ijms-26-10711-f004:**
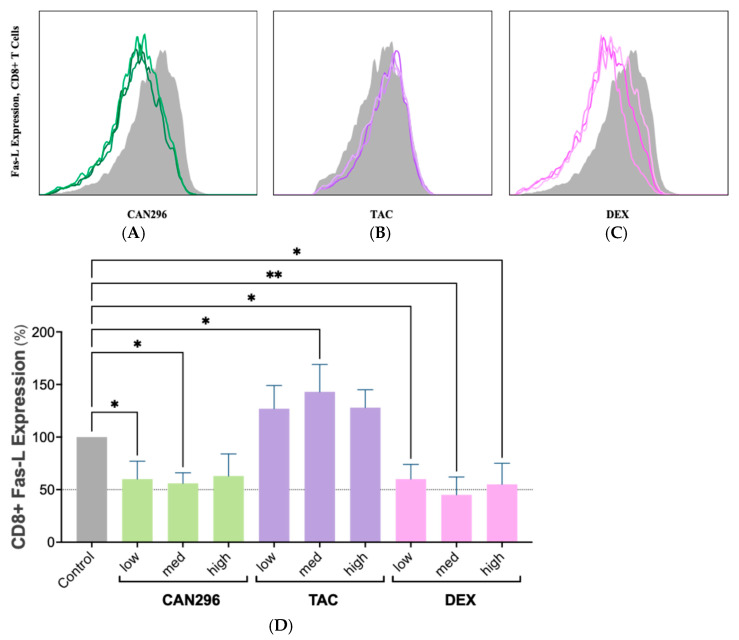
CAN296 reduced Fas-L expression in CD8^+^ T cells, while TAC increased expression at higher doses. (**A**–**C**) Representative histograms of Fas-L expression in CD8^+^ T cells (*n* = 2 donors) after 24 h treatment with CAN296 (2–8 µg/mL), TAC (0.1–10 ng/mL), or DEX (0.4–40 µg/mL). Darker shades indicate higher concentrations; gray lines represent positive controls (stimulated, untreated). (**D**) Quantified Fas-L levels (MFI, % of control). The dashed horizontal line marks 50% of the normalized control value, serving as visual reference thresholds for Fas-L expression reduction. CAN296 and DEX reduced Fas-L expression by ≈40–55%, whereas TAC elevated it at higher doses. Data are presented as medians ± 95% CIs. Statistical significance was determined using Šidák’s multiple comparisons test (* *p* < 0.05, ** *p* < 0.01). Abbreviations: CD8^+^, Cluster of Differentiation 8 cytotoxic T cells; Fas-L, Fas ligand; TAC, tacrolimus; DEX, dexamethasone; MFI, mean fluorescence intensity; CI, confidence interval.

**Figure 5 ijms-26-10711-f005:**
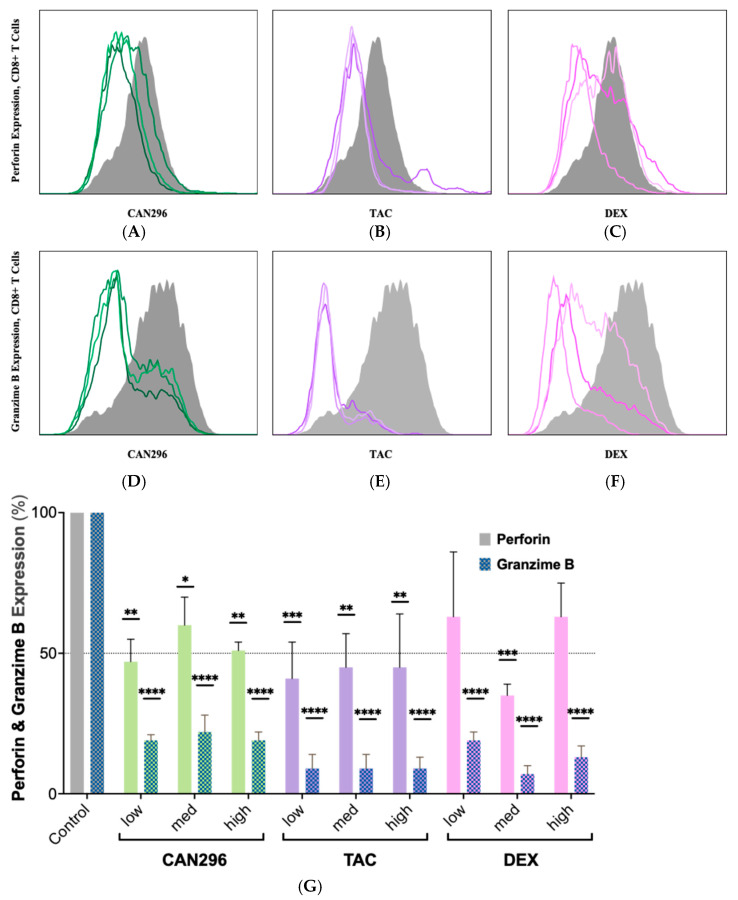
Effects of CAN296, TAC, and DEX on Perforin and Granzyme B expression in CD8^+^ T cells. (**A**–**C**) Representative histograms of Perforin and (**D**–**F**) Granzyme B expression in CD8^+^ T cells (*n* = 2 donors) treated with CAN296 (2–8 µg/mL), TAC (0.1–10 ng/mL), or DEX (0.4–40 µg/mL). Darker shades indicate higher concentrations; gray lines represent positive controls (stimulated, untreated). (**G**) Quantified Perforin and Granzyme B levels (MFI, % of control). The dashed horizontal line marks 50% of the normalized control value, serving as visual reference threshold for expression reduction. Data are presented as medians ± 95% CIs. Statistical significance was determined using Šidák’s multiple comparisons test (* *p* < 0.05, ** *p* < 0.01, *** *p* < 0.001, **** *p* < 0.0001). Abbreviations: CD8^+^, Cluster of Differentiation 8 cytotoxic T cells; TAC, tacrolimus; DEX, dexamethasone; MFI, mean fluorescence intensity; CI, confidence interval.

## Data Availability

The data presented in this study are available on request from the corresponding author. The data are not publicly available due to ethical restrictions related to donor confidentiality.

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
