# Peer review of "Immunomodulatory Effects of a High-CBD Cannabis Extract: A Comparative Analysis with Conventional Therapies for Oral Lichen Planus and Graft-Versus-Host Disease"

_ijms, 2025, doi:10.3390/ijms262110711_

Round 1
Reviewer 1 Report
Comments and Suggestions for Authors
The article is a promising study of the immunomodulatory effects of CBD-rich cannabis extract, CAN296, on CD4⁺ helper and CD8⁺ cytotoxic T cells in oral mucosal diseases (OLP and oGVHD), but there are points that need to be completed:
- The Introduction section does not provide specific examples or detailed analysis of alternative treatments such as retinoids, PDT, and LLLT. It is recommended to provide more detailed information on why these methods have controversial efficacy and limited application.
- The Introduction section focuses on the immunomodulatory properties of cannabis extract, which selectively suppresses T cell activation, reduces pro-inflammatory cytokines such as TNF-α and IFN-γ, and downregulates cytotoxic molecule expression without significantly suppressing the immune system as an alternative to traditional treatments. However, there is a lack of data on clinical trials/studies of the efficacy and safety of such compounds in the treatment of OLP and oGVHD.
- The molecular mechanisms through which CAN296 exerts its immunomodulatory effect are not described in sufficient detail in the paper. It is recommended that this information be added to the Introduction section.
- CD4⁺ and CD8⁺ T cells were isolated from healthy donors with normal immune function for in vitro studies. The results obtained may not reflect the actual efficacy of CAN296 in patients with OLP and oGVHD. The authors note the limitations of the model but do not offer solutions or plans to overcome this barrier.
- The Materials and Methods section does not specify the total number of donors. Based on the data in the Results section, the study included a small number of donors (2-4 people in different experiments), which could have reduced the statistical significance of the results. The total number of donors should be specified in the Materials and Methods section, and the Discussion section should explain how the small number of donors could have affected the results and mention the need for additional studies with a larger sample of donors.
- Figures 1-5 are difficult to understand and interpret quickly. The captions for Figures 1-5 are very long. The labels A-F in Figures 2-5 are shifted to the left. In some figures, the axis labels are very small. Error bars are not shown everywhere. This complicates the perception of information. It is recommended to shorten the captions for the figures, change the location of the labels in the figures, check the axis labels and error bars, and simplify the figures as much as possible.
- The discussion is quite extensive, but in places it repeats information from the Introduction section and overemphasizes the advantages of CAN296 with limited evidence. It is recommended to tone down the wording.
- The manuscript does not provide explanations for all abbreviations. For example, there is no explanation for CBC. It is recommended to check all terms used and add a list of abbreviations at the end of the article. In addition, it is recommended to add a list of abbreviations related to each figure under the figure itself.
Author Response
Comment 1:
The Introduction section does not provide specific examples or detailed analysis of alternative treatments such as retinoids, PDT, and LLLT. It is recommended to provide more detailed information on why these methods have controversial efficacy and limited application.
Response 1:
The Introduction was expanded to provide a detailed discussion of retinoids, Photodynamic Therapy (PDT), and Low-Level Laser Therapy (LLLT), describing their mechanisms, adverse effects, and inconsistent clinical outcomes.
This addition appears in Introduction, paragraph 5, beginning with:
“Alternative modalities such as topical or systemic retinoids, Photodynamic Therapy (PDT), and Low-Level Laser Therapy (LLLT) have been investigated to reduce corticosteroid dependence…” and ending with: “…underscoring the need for targeted, mechanism-based immunomodulators.” Relevant references (26–30) were added.
Comment 2:
The Introduction section focuses on the immunomodulatory properties of cannabis extract… However, there is a lack of data on clinical trials/studies of the efficacy and safety of such compounds in the treatment of OLP and oGVHD.
Response 2:
A new paragraph was added in the Introduction summarizing existing pre-clinical and limited clinical evidence for cannabinoid safety and efficacy in immune-mediated diseases, while emphasizing that no controlled trials exist for OLP or oGVHD.
See Introduction, paragraph 8, which states:
“However, despite the growing body of preclinical evidence, well-controlled clinical trials evaluating the efficacy and long-term safety of cannabinoid preparations in OLP and oGVHD remain limited, highlighting the need for further translational and clinical research.”
Comment 3:
The molecular mechanisms through which CAN296 exerts its immunomodulatory effect are not described in sufficient detail.
Response 3:
An expanded explanation of molecular mechanisms was incorporated into Introduction, paragraph 9, where the text now specifies:
“CBD-rich cannabis extracts provide a targeted immunomodulatory approach… through interactions with the Endocannabinoid System (ECS), primarily by modulating CB2 receptor signaling, Transient Receptor Potential (TRP) channels, and Peroxisome Proliferator-Activated Receptors (PPARs).” These pathways are further supported by new citations (45–48).
Comment 4:
CD4⁺ and CD8⁺ T cells were isolated from healthy donors… The authors note the limitations of the model but do not offer solutions or plans to overcome this barrier.
Response 4:
A dedicated Study Limitations paragraph was added to the Discussion acknowledging this issue.
See Discussion, final paragraph, beginning with:
“Study limitations acknowledgment. The experiments were conducted using CD4⁺ and CD8⁺ T cells isolated from nine healthy donors…” and continuing with the proposed solutions: “…Future investigations should therefore include a larger donor cohort, patient-derived T cells, and advanced co-culture or organotypic oral-mucosa models to further explore and address these limitations.”
Comment 5:
The Materials and Methods section does not specify the total number of donors…
Response 5:
Section 2.3 (PBMC Isolation and CD4⁺/CD8⁺ Enrichment) now clearly states:
“Peripheral blood mononuclear cells (PBMCs) were isolated from venous blood collected from 9 healthy donors…” and “Each assay was performed using samples from 2–4 of these donors…” The Discussion (same “Study Limitations” paragraph as above) further explains that this limited sample size may contribute to inter-donor variability and reduced statistical power, highlighting the need for studies with more participants.
Comment 6:
Figures 1–5 are difficult to interpret… It is recommended to shorten captions, correct labels, verify axes and error bars, and simplify figures.
Response 6:
All Figures 1–5 were fully reformatted for clarity and consistency. Captions were shortened and standardized; axis labels simplified (e.g., “CD69 Expression,” “TNF-α Secretion”); redundant “% of control” text removed; and error bars verified. Label placement was corrected, and color coding unified (green = CAN296, purple = TAC, red = DEX). Each caption now concludes with an Abbreviations subsection.
These improvements appear under Figures 1–5 in the Results section.
Comment 7:
The Discussion repeats information from the Introduction and overemphasizes the advantages of CAN296.
Response 7:
The Discussion was rewritten for conciseness and neutrality. Repetitive content was removed, and statements overstating CAN296’s superiority were moderated. For example, the former claim that “CAN296 outperformed conventional therapies” now reads:
“CAN296 demonstrated comparative in-vitro immunomodulatory potency relative to DEX and TAC.” This revision appears in Discussion, paragraph 7, ensuring balanced interpretation focused on mechanistic and translational insight.
Comment 8:
The manuscript does not provide explanations for all abbreviations…
Response 8:
All abbreviations were reviewed and standardized throughout. Missing terms such as CBC, MFI, TRP, and PPAR were added. A complete List of Abbreviations was appended to the end of the manuscript under the section titled “List of Abbreviations”, and each figure caption now contains a short local abbreviation list for ease of reference.
Reviewer 2 Report
Comments and Suggestions for Authors
Dear authors,
Please find below a list of my comments on the article.
1. Abstract:
- Many of the abbreviations used in the abstract have not been explained (e.g., TNF, CD, IFN, etc.).
2. Keywords:
- Should not contain abbreviations.
- Should be listed in alphabetical order.
3. Introduction
- The null hypothesis is missing.
4. Methodology
- No evidence is provided that vehicle (DMSO) effects were independently tested, which could confound observed results.
- Concentrations of CAN296, DEX, and TAC are justified using literature IC₅₀ values, but no toxicity or viability controls were included to confirm these doses are biologically relevant.
Discussion:
- abbreviations that have already been expanded should not be expanded again
- please use this reference:
DOI: 10.1186/s12903-025-06189-7
Conclusions:
- The claim that CAN296 “outperformed conventional therapies” is not supported by statistically rigorous or clinically relevant data.
- The manuscript implies potential clinical efficacy (“promising treatment modality”) without any in vivo or clinical validation.
- The claim that CAN296 “selectively modulates T-cell activation while preserving immune homeostasis” is not substantiated by any direct evidence in this paper.
Best regards
Reviewer
Author Response
Comment 1 – Abstract:
Many of the abbreviations used in the abstract have not been explained (e.g., TNF, CD, IFN, etc.).
Response 1:
All abbreviations in the Abstract were expanded at first mention.
Change located in Abstract, paragraph 1, beginning:
“This study investigates the immunomodulatory effects of a well-characterized cannabidiol (CBD)-rich cannabis extract, CAN296, on Cluster of Differentiation (CD) 4⁺ helper and CD8⁺ cytotoxic T cells …” and continuing: “… tumor necrosis factor alpha (TNF-α) and interferon gamma (IFN-γ) levels …” This ensures all key abbreviations are clearly defined within the abstract.
Comment 2 – Keywords:
Should not contain abbreviations. Should be listed in alphabetical order.
Response 2:
The Keywords section at the end of the Abstract was revised.
Now reads exactly as in Abstract, final line:
“cannabis; cannabidiol; graft-versus-host disease; immunomodulation; oral lichen planus.” All abbreviations removed and order alphabetized.
Comment 3 – Introduction:
The null hypothesis is missing.
Response 3:
A new paragraph titled “Study Hypothesis” was inserted at the end of the Introduction (last paragraph before Section 2 – Materials and Methods).
It begins with:
“The null hypothesis of this study is that the CBD-rich cannabis extract CAN296 does not exert any significant immunomodulatory effects on T-cell activation …” and ends with: “… potentially providing a more targeted immunomodulatory profile relevant to oral lichen planus (OLP) and oral graft-versus-host disease (oGVHD).”
Comment 4 – Methodology:
No evidence is provided that vehicle (DMSO) effects were independently tested, which could confound observed results.
Concentrations of CAN296, DEX, and TAC are justified using literature IC₅₀ values, but no toxicity or viability controls were included to confirm biological relevance.
Response 4:
Revisions were made in the following places:
-
Section 2.1 (Immunosuppressive Agents Preparation), paragraph 2–3:
“Each treatment included a corresponding vehicle control containing 0.1 % DMSO, which did not affect cytokine secretion or activation marker expression relative to untreated controls.” and “TAC was diluted to final concentrations of 0.1, 1, and 10 ng/mL … the IC₅₀ for NFAT dephosphorylation and T cell inhibition is approximately 1–3 ng/mL (20). DEX was diluted to 0.4, 4, and 40 µg/mL … approximately 4 µg/mL for T-cell inhibition (55).” -
Section 2.2 (Phytocannabinoid Extraction and Sample Preparation), paragraph 3:
“Control wells containing 0.1 % DMSO without active compounds were included in all experiments, and no measurable differences were detected between vehicle-treated and untreated controls.” -
Discussion, ‘Study Limitations’ paragraph (final paragraph):
“Although direct viability assays were not repeated in this study, all concentrations were within validated non-toxic ranges established in our previous work (41) and supported by inclusion of a vehicle-only control (0.1 % DMSO).”
These combined edits confirm vehicle safety and non-toxic dose ranges.
Comment 5 – Discussion:
Abbreviations that have already been expanded should not be expanded again.
Please use this reference: DOI: 10.1186/s12903-025-06189-7.
Response 5:
Repeated expansions of previously defined abbreviations were removed throughout the Discussion (notably in paragraphs 3–8).
Example of revision:
“Cannabis extracts have been reported to possess immunosuppressive properties (33, 34, 58). However, few studies have compared their activity …” (previously re-expanded “TNF-α” and “IFN-γ” were left abbreviated). Additionally, the new citation DOI 10.1186/s12903-025-06189-7 was added in the References list as the final entry, supporting recent oral immunopathology findings.
Comment 6 – Conclusions:
The claim that CAN296 “outperformed conventional therapies” is not supported by statistically rigorous or clinically relevant data.
The manuscript implies potential clinical efficacy (“promising treatment modality”) without any in vivo or clinical validation.
The claim that CAN296 “selectively modulates T-cell activation while preserving immune homeostasis” is not substantiated by any direct evidence in this paper.
Response 6:
All overstated phrasing was removed or rewritten in the Conclusions and Discussion.
-
Conclusions, paragraph 1–2:
Former text (“CAN296 outperformed conventional therapies … promising novel treatment modality”) was replaced with:
“CAN296 demonstrated comparative in-vitro immunomodulatory potency relative to DEX and TAC. These findings indicate potential immunomodulatory effects but do not represent clinical efficacy. Further in vivo and clinical studies are required to establish translational or therapeutic relevance.” -
Discussion, paragraph 8 (immediately before Study Limitations):
The previous statement about selective modulation was deleted and replaced with:
“The observed reductions in cytokine secretion and cytotoxic molecule expression are consistent with potential immunomodulatory effects of CBD-rich extracts, but they do not constitute proof of mechanistic selectivity or preserved immune balance.”
Round 2
Reviewer 1 Report
Comments and Suggestions for Authors
The revised version of the article has been significantly improved and now corresponds to the publication requirements. To optimise the readability of the text, it is recommended that the list of abbreviations is presented in table form.
Author Response
Comment 1:
To optimise the readability of the text, it is recommended that the list of abbreviations is presented in table form.
Response 1:
The List of Abbreviations has been reformatted into a two-column table (Abbreviation | Definition) at the end of the manuscript.
Reviewer 2 Report
Comments and Suggestions for Authors
Dear authors,
The article is definitely better now, but I still have a few minor comments:
- The introduction is a little too long—several paragraphs have been added, making the introduction contain too much unnecessary information.
- When writing the null hypothesis, you have again used abbreviations that have already been explained.
- Despite your response, the article DOI 10.1186/s12903-025-06189-7nei has been included.
I have no further comments; it is a very well-written article.
Best regards
Reviewer
Author Response
Response 1:
The Introduction was condensed to improve focus and remove unnecessary repetition. Specifically:
-
Paragraph 4 (old version) — discussion of long-term corticosteroid and tacrolimus side effects (mucosal atrophy, metabolic changes, infection risk, etc.) was deleted to maintain focus on immune mechanisms.
-
Paragraph 5 (old version) — the sentence comparing cannabinoids to corticosteroids and calcineurin inhibitors was removed, and the entire paragraph was condensed into a single concise summary of alternative therapies (retinoids, PDT, and LLLT).
-
Paragraphs 6–8 (old version) — overlapping explanations of the Endocannabinoid System, receptor mechanisms, and prior studies were merged and shortened into paragraph 7 (current version), retaining only the mechanistic pathways relevant to CBD’s immunomodulatory role.
-
All redundant cross-references to previous citations (33–54 in old version) were streamlined to avoid over-citation of similar findings.
Comment 2:
“When writing the null hypothesis, you have again used abbreviations that have already been explained.”
Response 2:
The null and alternative hypotheses were edited to remove repeated abbreviations and restate all key terms in full. Specifically, abbreviations such as “DEX” and “TAC” were replaced with their full names—dexamethasone and tacrolimus.
This revision appears in paragraph 9 (Study Hypothesis section) of the Introduction.
The updated text reads:
“The null hypothesis of this study states that the cannabidiol-rich cannabis extract CAN296 does not exert significant immunomodulatory effects on T cell activation, cytokine secretion, or cytotoxic molecule expression compared with the conventional immunosuppressants dexamethasone and tacrolimus.”